# Effects of Soluble Organic Fertilizer Combined with Inorganic Fertilizer on Greenhouse Tomatoes with Different Irrigation Techniques

Binnan Li and Lixia Shen *

College of Water Conservancy and Engineering, Taiyuan University of Technology, Taiyuan 030024, China; libinnan0078@link.tyut.edu.cn
* Correspondence: shenlixia@tyut.edu.cn

**Abstract:** A reasonable fertilization rate and appropriate irrigation technology can lead to the green development of greenhouse tomatoes. The purpose of this study was to explore suitable irrigation technology for greenhouse tomatoes and the appropriate application rate of the soluble organic fertilizer and inorganic fertilizer combination. In 2021 and 2022, the effects of different irrigation techniques and fertilization treatments on tomato plant growth, fruit quality, yield, and efficiency were studied. The irrigation techniques in this study were drip and Moistube irrigation, and there were seven types of fertilization treatment, namely, no fertilization treatment (CK); low-volume (T1, 330 kg/hm$^2$), medium-volume (T2, 660 kg/hm$^2$), and high-volume inorganic fertilizer treatments (T3, 990 kg/hm$^2$); and three inorganic fertilizer treatments of low-volume inorganic fertilizer (T1, 330 kg/hm$^2$) combined with low-volume (F1, T1 + 75 kg/hm$^2$), medium-volume (F2, T1 + 225 kg/hm$^2$), and high-volume (F3, T1 + 375 kg/hm$^2$) organic fertilizer. A total of 14 experimental treatments were implemented for irrigation and fertilization. The results of the two-year experiment show that the growth effect on the height, stem diameter, and leaf area index of tomato plants was the best using the treatment of low-concentration inorganic fertilizer combined with medium-concentration organic fertilizer with Moistube irrigation and drip irrigation. Using the two irrigation methods, the application of soluble organic fertilizer increased the yield and improved the fruit quality of the tomato. The maximum yield increased by 28.52%, the soluble sugar content increased by 14.49%, the vitamin C content increased by 45.04%, and the lycopene increased by 18.79%. The entropy-weight TOPSIS model was used to comprehensively evaluate 14 evaluation objects with different irrigation methods and fertilization treatments. The results of the two-year experiment show that the best fertilization treatment under Moistube irrigation and drip irrigation conditions was low-concentration inorganic fertilizer combined with medium-concentration soluble organic fertilizer, which was combined with the best fertilization treatment, and the most suitable irrigation method for greenhouse tomato cultivation in the Loess Plateau was Moistube irrigation. The results of this study also provide practical experience and theoretical support for adaptive irrigation and the integrated management of water and fertilizer.

**Keywords:** greenhouse tomato; combined application of organic fertilizer and inorganic fertilizer; Moistube irrigation; drip irrigation; TOPSIS





## 1. Introduction

Tomatoes are rich in nutritional value, containing vitamins, minerals, carotenoids, lycopene, and other nutrients [1,2]. These ingredients are beneficial to human health, such as skin protection and anti-aging, as well as the prevention and treatment of hypertension and cardiovascular diseases [3]. Tomatoes also have a high economic value because, in addition to their daily dietary uses, they can be made into and sold as tomato juice and tomato sauce. Tomatoes have become the most valued and widely consumed vegetable [4].

Tomato varieties are constantly being enriched, and the cultivation area is continuously being expanded. In order to harvest higher tomato yields and economic benefits, the overuse of chemical fertilizers is becoming more common, which leads to a range of soil and environmental problems, for example, declining soil fertility, soil nutrient imbalance, soil compaction, reduction in soil organic matter, and pollution of the soil environment [5,6]. Organic fertilizer contains a variety of trace elements as well as organic matter, and it has a long shelf life and long-lasting fertility. Organic fertilizer can promote microbial reproduction, improve the microbiota of the crop root zone, and improve the resistance of plants to diseases and insects [7,8]. However, traditional organic fertilizers mostly comprise chicken manure, grass charcoal, biogas fertilizer, etc., as in the basic application and are applied to the field at one time, which involves a large workload. Soluble organic fertilizer can be used as a top dressing and applied to the field with irrigation water. It significantly increases the integration of fertilizer and irrigation water efficiency. Relevant studies [9–11] have shown that the combined application of soluble organic fertilizer and inorganic fertilizer has a better effect than the sole application. The combined application can improve fertilizer efficiency, increase crop nutrition, increase soil organic matter content, improve soil physical and chemical properties, and adjust soil pH [12–14]. Wen et al. [15] discussed the effect of chemical fertilizer reduction and the application of soluble organic fertilizer on processed tomatoes and concluded that the combined application of organic fertilizer and inorganic fertilizer could boost tomato yields, enhance fruit quality, and create a healthy soil microclimate. Wu et al. [16] analyzed the effects of drip irrigation of tomato with different irrigation levels and fertilizers, and the results showed that the combined application of soluble organic fertilizer and inorganic fertilizer achieved higher profit and water use efficiency and significantly improved the yield of tomato. Qi et al. [17] studied the effects of the combined application of organic and inorganic fertilizers on the nutrients, yield, and quality of the soil tillage layer of tomatoes, and the results showed that the combined application of soluble organic fertilizers with inorganic fertilizer reduction could significantly improve the physical and chemical properties of soil and improve the yield and quality of tomatoes. The above studies showed that soluble organic fertilizer is widely used in tomato cultivation, and the application of inorganic fertilizer in combination with inorganic fertilizer could increase tomato yield, improve fruit quality, and improve the soil environment. At present, the influence of soluble organic fertilizer combined with inorganic fertilizer on greenhouse tomatoes in the Loess Plateau with different irrigation methods has not been sufficiently studied.

A novel kind of underground water-saving irrigation technique called Moistube irrigation has been developed. The core technology is a semi-permeable membrane with unidirectional permeability, which can slowly disperse water into the soil. Soil water flow during irrigation is comparable to linear source infiltration, which includes both upward and horizontal diffusion in addition to downward infiltration, so that it can provide uniform and continuous irrigation water for crops [18–20]. Relevant studies have reported that Moistube irrigation technology can improve the stomatal conductance of crop leaves, improve water use efficiency, promote crop growth [21,22], achieve good water-saving and yield increases, improve the soil environment [16,23,24], and have great development potential. In addition, drip irrigation, as one of the most effective water-saving irrigation technologies, is suitable for the irrigation of fruit trees, vegetables, cash crops, and greenhouses. Many studies have shown that drip irrigation can significantly increase tomato dry matter accumulation and yield compared with traditional irrigation methods, and when used in combination with soluble fertilizers, it can not only promote the uptake of tomato nutrients but also significantly improve fertilizer use efficiency [5,25]. At present, the cultivation of facility tomatoes under Moistube irrigation conditions and the rational application of soluble organic fertilizer in the Loess Plateau have not been explored in sufficient detail. In addition, the selection of suitable water-saving technology for the cultivation of greenhouse tomatoes in this area in combination with the application of soluble organic fertilizer remains to be further studied and discussed.

A reasonable fertilization rate and appropriate irrigation method can achieve the green development of greenhouse tomatoes. Therefore, the purpose of this study was to explore the growth law and influence mechanism of inorganic fertilizer combined with soluble organic fertilizer on the growth of greenhouse tomatoes in the Loess Plateau and to determine the most appropriate combined application rate according to the quantitative description of tomato yield, quality, and water and fertilizer use efficiency. At the same time, the entropy-weight method TOPSIS model was used to comprehensively evaluate the four aspects of tomato plant growth, quality, yield, and efficiency according to the effects of Moistube irrigation and drip irrigation with different fertilization treatments, and the most suitable irrigation technology for greenhouse tomato cultivation in this area was explored.

## 2. Materials and Methods

### 2.1. Experimental Location and Materials

The trial was carried out from May to October in 2021 and 2022 at the tomato industrial park in Liujiabao Township, Xiaodian District, Taiyuan City, Shanxi Province. The greenhouse had a length of 50 m, a span of 14 m, and a maximum height of 4 m, and the shed was covered with transparent polyethylene plastic film and roller carpet. The experimental area belonged to a temperate continental monsoon climate with a large temperature difference between day and night and the characteristics of four distinct seasons. The annual average temperature was 11 °C, the annual precipitation was 520 mm, and the average amount of sunshine was 2672 h. The soil in the experimental area was sandy loam, and its sand, silt, and clay contents were 64.54%, 25.42%, and 10.04%, respectively. More information on the physicochemical properties of the soil is shown in Table 1. The test material was "Provencal" pink tomatoes. Details of the meteorological data in the greenhouse for the two-year experiment are shown in Figure 1.

**Table 1.** Physicochemical properties of the soil in the experimental area.

| Soil Physicochemical Properties | Data |
| --- | --- |
| Soil type | Sandy loam |
| Soil pH | 7.2 |
| Bulk density (g/cm$^3$) | 1.54 |
| Field capacity (%) | 26.71 |
| Soil nitrogen content (mg/kg) | 12.30 |
| Available phosphorus content (mg/kg) | 19.23 |
| Available potassium content (mg/kg) | 116.60 |
| Organic matter content (g/kg) | 11.23 |

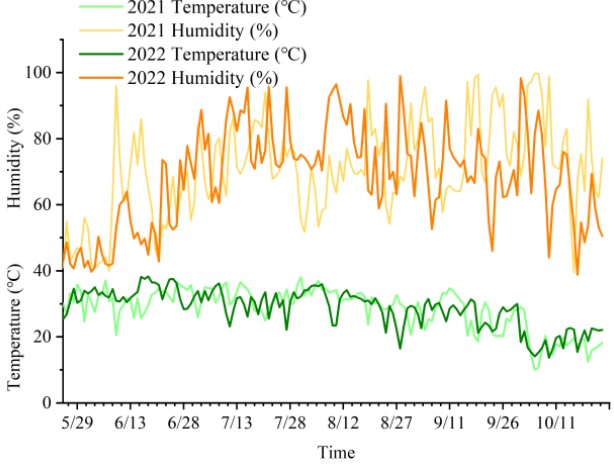

**Figure 1.** Greenhouse meteorological data for 2021 and 2022.

## 2.2. Experimental Design

In this experiment, no fertilization was used in the control group, and seven fertilization treatments were designed with reference to the conventional fertilization amounts of local farmers (N/P/K = 300:120:240): CK, no fertilization; T1, low-concentration inorganic fertilizer; T2, medium-concentration inorganic fertilizer; T3, high-concentration inorganic fertilizer; F1, low-concentration inorganic fertilizer + low-concentration soluble organic fertilizer; F2, low-concentration inorganic fertilizer + medium-concentration soluble organic fertilizer; F3: low-concentration inorganic fertilizer + high-concentration soluble organic fertilizer. Each treatment was repeated 3 times. Inorganic fertilizers used were urea (N46%), potassium dihydrogen phosphate ($P_2O_5$ 52%, $K_2O$ 33.8%), and potassium sulfate ($K_2O$ 52%). Organic fertilizers were commercially soluble organic fertilizers (N 13%, $P_2O_5$ 5%, $K_2O$ 13%, organic matter content ≥40%). The specific fertilizer dosage of each treatment with the two irrigation methods used in the experiment is shown in Table 2.

**Table 2.** The amount of fertilizer under Moistube irrigation and drip irrigation conditions.

| Treatments | | N ($kg/hm^2$) | $P_2O_5$ ($kg/hm^2$) | $K_2O$ ($kg/hm^2$) | Organic Fertilizer ($kg/hm^2$) |
|---|---|---|---|---|---|
| Moistube Irrigation | Drip Irrigation | | | | |
| MCK | DCK | 0 | 0 | 0 | 0 |
| MT1 | DT1 | 150 | 60 | 120 | 0 |
| MT2 | DT2 | 300 | 120 | 240 | 0 |
| MT3 | DT3 | 450 | 180 | 360 | 0 |
| MF1 | DF1 | 150 | 60 | 120 | 75 |
| MF2 | DF2 | 150 | 60 | 120 | 225 |
| MF3 | DF3 | 150 | 60 | 120 | 375 |

Note: N fertilizer is urea, $P_2O_5$ uses potassium dihydrogen phosphate, and $K_2O$ uses potassium. sulfate. CK is no fertilization; T1, T2, and T3 are low-, medium-, and high-concentration inorganic fertilizers, respectively; and F1, F2, and F3 are the combinations of low-, medium-, and high-concentration soluble organic fertilizers and low-concentration inorganic fertilizers. M and D stand for Moistube irrigation and drip irrigation, respectively.

In the experiment, the irrigation methods were Moistube irrigation and drip irrigation. The irrigation amount was obtained from the statistics of the water consumption of the water tank, the pressure head was set to 2 m, the Moistube was a ridge and two pipes, the buried depth was 15 cm, and the Moistube spacing was 30 cm. Irrigation was provided during the whole growth period of the tomato plant. Drip irrigation was carried out using a ridge with two pipes, and the pipe spacing was 25 cm. The total amount of irrigation water commonly used in the local area was 250–320 mm. The frequency of irrigation during the tomato growth period was once every 7–10 days. Based on this reference and according to the relevant reports of drip irrigation water-saving parameters [22,23], the upper limit of drip irrigation water in this experiment was set to 90% of the field water-holding capacity, the irrigation quota was calculated from the irrigation amount formula, the irrigation quota was 25.64 mm, and the amount of irrigation water was the same each time. The irrigation frequency was set at 7 days/time at the seedling stage and 10 days/time from the flowering stage to the fruiting stage, and a total of 8 periods of irrigation of 205 mm were applied during the whole tomato growth period. The irrigation amount formula [22] is as follows:

$$I = (0.9\theta_{Fc} - \theta_v) \times Z_r \times S \times p$$

In this paper, $I$ is the irrigation amount of each growth period, $cm^3$; $\theta_{Fc}$ is the field water holding capacity; $\theta_v$ is the soil water content before irrigation, $cm^3/cm^3$; $Z_r$ is the planned wetting layer depth, 0.6 m; $S$ is the irrigation area of each treatment, $m^2$, and $p$ is the wetting ratio; 0.6 was taken in this study. In this trial, irrigation water was used from deep underground water in the area. The field layout was such that the width of the ridges was 60 cm, the height was 15 cm, the spacing was 50 cm, and the length was 9 m. In this study, a total of 14 treatments were performed, and each treatment was repeated 3 times. The experimental field area of each treatment was 35.1 $m^2$. Tomato plants were transplanted with five leaves and one heart and planted in two rows on one ridge with a

plant spacing of 30 cm and a planting density of 30,744 plants/hm$^2$. At the time of planting, the height of the plant was about 15 cm, and the stem thickness was about 5 mm. All field management was consistent during the tomato growth period. The fertilizer was fully dissolved according to the set concentration ratio and then applied to the field with water, and the fertilizer was applied as top dressing. A total of 5 periods of fertilization were applied during the whole tomato growth period, and the fertilizer was allocated according to the fertilizer demand of the tomato in each growth period [24], once at the seedling stage and once at the flowering and fruit setting stage. A total of 12.5% of the total amount of fertilizer was applied. From the fruit expansion period to the harvest stage, 25% of the total amount of fertilizer was applied a total of 3 times. The number of days of each tomato growth period is shown in Table 3.

**Table 3.** The number of days of each tomato growth period.

| Irrigation Method | Year | Number of Days during the Whole Growth Period (d) | | |
| --- | --- | --- | --- | --- |
| | | Seeding Stage | Flowering Stage | Fruiting Stage |
| Moistube irrigation | 2021 | 39 | 21 | 75 |
| | 2022 | 38 | 22 | 73 |
| Drip irrigation | 2021 | 40 | 26 | 76 |
| | 2022 | 39 | 25 | 74 |

*2.3. Measurement Indicators and Methods*

2.3.1. Growth Indicators

To determine the vertical height of the tomato plant above the ground surface, the plant height was recorded using a tape measure. The stem diameter was measured using vernier calipers at the upper, middle, and lower parts of the stem and then averaged. The leaf area index was weighed using the paper-cut weighing method [25,26]. After sampling and weighing the leaves of the plant, 1/50 of the leaves are taken and drawn on standard graph paper, and then the paper mold is cut out and weighed, and the leaf area index per plant is calculated as follows:

$$LAI = AR \times (PW/SPW) \times 50 \times d/10^6 \qquad (1)$$

where LAI is leaf area index, AR is the area of standard graph paper, PW is the weight of the paper mold, SPW is the weight of standard graph paper, and d is the planting density in plants/m$^2$.

2.3.2. Quality Indicators

When the tomatoes were ripe, fresh fruits were picked for quality determination. The measurement indicators and methods are shown in Table 4. Total soluble sugars were determined using a handheld refractometer (RHBO-90) [26]. Vitamin C was titrated with sodium 2,6-dichloroindigophenol [27]. Fresh samples were added to an oxalic acid solution, ground, filtered, and titrated with sodium 2,6-dichloroindifol. Finally, the vitamin C content was calculated. Lycopene was subjected to ultraviolet spectrophotometry [28]. The standard curve was drawn with the Sudan I pigment standard solution, and the red pigment of the test solution was extracted using methanol and toluene. The absorbance value was then measured using a UV spectrophotometer (T6-1650E, Beijing PERSEE, Beijing, China) and incorporated into the formula of the standard curve to calculate the lycopene content. Nitrate was also measured using ultraviolet spectrophotometry [29]. First, potassium nitrate solution was used to produce a standard curve, and salicylic-acid-concentrated sulfuric acid and sodium hydroxide solution were added to the sample solution to be tested. After a sufficient reaction, the absorbance value was determined using a UV spectrophotometer and then incorporated into the formula of the standard curve to calculate the nitrate content. Titratable acids were titrated using the NaOH neutralization

titration method [30], and the sugar/acid ratio was calculated using the ratio of soluble sugar to titratable acid [31].

**Table 4.** Method for determining tomato quality.

| Number | Quality Indicator | Measurement Method |
|--------|-------------------|---------------------|
| 1 | Total Soluble Sugars (TSU) | Handheld refractometer |
| 2 | Nitrate Content (NC) | Ultraviolet spectrophotometer |
| 3 | Titratable Acids (TA) | Neutralization titration |
| 4 | Vitamin C Content (VC) | Titration method |
| 5 | Lycopene Content (LYC) | Ultraviolet spectrophotometer |
| 6 | Sugar/Acid Ratio (SAR) | Sugar divided by acid |

2.3.3. Production Indicators

After the tomato fruit ripened, 3 plants were randomly selected for each treatment, and 5 tomato fruits were picked per panicle to determine the average fruit weight and fruit number per plant. After taking the three spikes, the yield of different fertilization treatments was calculated according to the measurement results of the electronic scale, and each treatment was repeated 3 times.

2.3.4. Water and Fertilizer Use Efficiency

The irrigation water use efficiency is calculated as follows:

$$WUE = Y/ET \tag{2}$$

$$ET = P + W + K - R - L - (T_0 - T_t) \tag{3}$$

$$ET = W - (T_0 - T_t) \tag{4}$$

Among them, *ET* calculates the water consumption of tomatoes during the whole growth period according to the water balance method [29]. The calculation formula is shown in (3), where *ET* is the water consumption, mm; *P* is effective rainfall, mm; *W* is the amount of irrigation water, mm; *K* is the amount of groundwater recharge in the time period *t*, mm; *R* is the runoff, mm; *L* is the amount of deep leakage, mm; and $T_0$ and $T_t$ are the water storage in the planned wet layer of the soil at the beginning of the period and at any time *t*, mm. According to actual measurements, the soil moisture at a depth of 60 cm during the growth period did not change substantially, and tomatoes were grown in greenhouses, so *P*, *K*, *R*, and *L* were negligible. Equation (3) is simplified to Equation (4).

The formula for the partial fertilizer productivity is:

$$PFP = Y/F \tag{5}$$

where PFP is the partial fertilizer productivity, kg/kg; Y is the tomato yield, kg/hm$^2$; and F is the total amount of N, $P_2O_5$, and $K_2O$ input, kg/hm$^2$.

*2.4. TOPSIS Comprehensive Evaluation Model of the Entropy-Weight Method*

2.4.1. The Entropy-Weight Method Determines the Weight of Each Index

When determining the weights of each index, the entropy-weight method can reduce the bias caused by subjective factors and can modify the determined weights to make the results more consistent with the actual situation. Therefore, the entropy-weight method was used to calculate the weight of each index according to the following steps:

(1)  Initial matrix

$$R = \left[G_{ij}\right]_{m \times n} \tag{6}$$

where *i* = 1, 2, 3, . . . *m*; *j* = 1, 2, 3, . . . *n*.

(2)   The processing of indicator data was standardized. In order to avoid the influence of the inconsistency of different index dimensions, the range method was used to standardize the data of each index. The normalization of positive and negative indicators is shown by the following formula:

$$r_{ij} = \frac{x_{ij} - x_{min}}{x_{max} - x_{min}} \tag{7}$$

$$r_{ij} = \frac{x_{max} - x_{ij}}{x_{max} - x_{min}} \tag{8}$$

(3)   Initial matrix R normalization

$$P = \left[P_{ij}\right]_{m \times n} \tag{9}$$

$$P_{ij} = r_{ij} / \sum_{j=1}^{n} r_{ij} \tag{10}$$

where $i = 1, 2, 3, \ldots m$; $j = 1, 2, 3, \ldots n$.

(4)   The entropy value and utility value of index information were calculated.

$$e_i = \left(-\sum_{j=1}^{n} P_{ij} \ln P_{ij}\right) / \ln n \tag{11}$$

$$d_i = 1 - e_i \tag{12}$$

where $e_i$ is the information entropy and $d_i$ is the utility value. When $P_{ij} = 0$, $\ln P_{ij}$ is meaningless. The following formula was used to calculate the calculation:

$$P'_{ij} = \left(1 + P_{ij}\right) / \sum_{j=1}^{n} \left(1 + P_{ij}\right) \tag{13}$$

(5)   The weight of the $i$th indicator was calculated.

$$u_i = (1 + e_i) / \sum_{i=1}^{m} (1 - e_i) \tag{14}$$

where $u_i$ is the indicator weight.

### 2.4.2. TOPSIS Model

TOPSIS is an analytical method that is suitable for multiple indicators and the comparative selection of multiple protocols [32,33]. In this method, the optimal scheme and the worst scheme of each index are determined, then the Euclidean distance between each scheme and the positive and negative ideal solutions is obtained, the proximity between each scheme and the optimal scheme is obtained, and the optimal scheme is selected according to the ranking.

(1)   Isonomialization of all indicators. When $G_{ij}$ is a high-performance indicator, $G'_{ij} = G_{ij}$; when $G_{ij}$ is a low-performance indicator, $G'_{ij} = 1/G_{ij}$.
(2)   A normalized initial matrix is constructed.

$$Z_{ij} = \frac{G_{ij}}{\sqrt{\sum_{i=1}^{m} x_{ij}^2}} \tag{15}$$

$$Z = \left[Z_{ij}\right]_{m \times n} \tag{16}$$

(3)   The positive and negative ideal solutions are determined. Based on the normalized matrix Z, the cosine method is used to find the positive ideal solution and the negative ideal solution. The positive ideal solution $Z^+$ is composed of the maximum value of

each column element, and the negative ideal solution $Z^-$ is composed of the minimum value of each column element, as follows:

$$Z^+ = \left(max\{z_{11}, z_{21}, \cdots, z_{i1}\}, max\{z_{12}, z_{22}, \cdots, z_{i2}\}, \cdots, max\{z_{1j}, z_{2j}, \cdots, z_{ij}\}\right)$$
$$Z^+ = \left(Z_1^+, Z_2^+, \cdots, Z_j^+\right) \tag{17}$$

$$Z^- = \left(min\{z_{11}, z_{21}, \cdots, z_{i1}\}, min\{z_{12}, z_{22}, \cdots, z_{i2}\}, \cdots, min\{z_{1j}, z_{2j}, \cdots, z_{ij}\}\right)$$
$$Z^- = \left(Z_1^-, Z_2^-, \cdots, Z_j^-\right) \tag{18}$$

(4) The Euclidean distance between each evaluation object is calculated along with the positive and negative ideal solutions.

$$D_i^+ = \sqrt{\sum_j \left(z_{ij} - Z_j^+\right)^2} \tag{19}$$

$$D_i^- = \sqrt{\sum_j \left(z_{ij} - Z_j^-\right)^2} \tag{20}$$

(5) The proximity of each evaluation object is calculated.

$$C_i = \frac{D_i^-}{D_i^- + D_i^+} \tag{21}$$

where $0 \leq C_i \leq 1$; within this range, the higher the $C_i$ value, the better the evaluation object.

### 2.5. Data Analysis Methods

The trial design was completely randomized. The significance analysis was carried out via one-way ANOVA analysis and multiple comparisons of LSD, along with hoc comparisons in SPSS 24 software. The entropy-weight method TOPSIS analysis was performed with METLAB 2014b, and the plot was created using Origin 2021 software.

### 3. Results

### 3.1. The Result of the Growth of Tomato Plants

Tomato growth in 2021 and 2022 is shown in Figure 2 (different lowercase letters in the figure indicate significant differences between treatments, $p < 0.05$). Using the same water-saving method, there was no significant difference between MT3 and MF2 or DT3 and DF2 in the three growth indices of plant height, stem diameter, and leaf area index, but compared with the control group, plant height, stem diameter, and leaf area index were significantly increased. Plant height, stem diameter, and leaf area index together showed an increasing trend with the concentration of inorganic fertilizer. Compared with the control group, the growth indices of MT3 treatment under Moistube irrigation conditions were the largest, with an average increase of 8.67%, 14.93%, and 30.66% in two years. The growth indices of DT3 treatment under drip irrigation conditions were the largest, with an average increase of 6.27%, 11.80%, and 33.97% in two years. The plant height, stem diameter, and leaf area of MT3 increased by 3.19%, 7.48%, and 7.49%, respectively, compared with DT3. Plant height, stem diameter, and leaf area index all showed a trend of initially growing and then dropping with the rise in the concentration of low-concentration inorganic fertilizer mixed with soluble organic fertilizer. The growth indices of MF2 treatment were the largest, with an average increase of 7.80%, 13.88%, and 27.68% in two years. The DF3 treatment had the largest growth indices under drip irrigation conditions, with an average increase of 5.60%, 10.89%, and 29.86% in two years. In addition, the plant height, stem diameter, and leaf area of MF2 increased by 3.02%, 7.35%, and 6.49%, respectively, compared with DF2. With the same fertilization treatment, the three growth indices of tomatoes using the Moistube irrigation method increased more than those using the drip irrigation method.

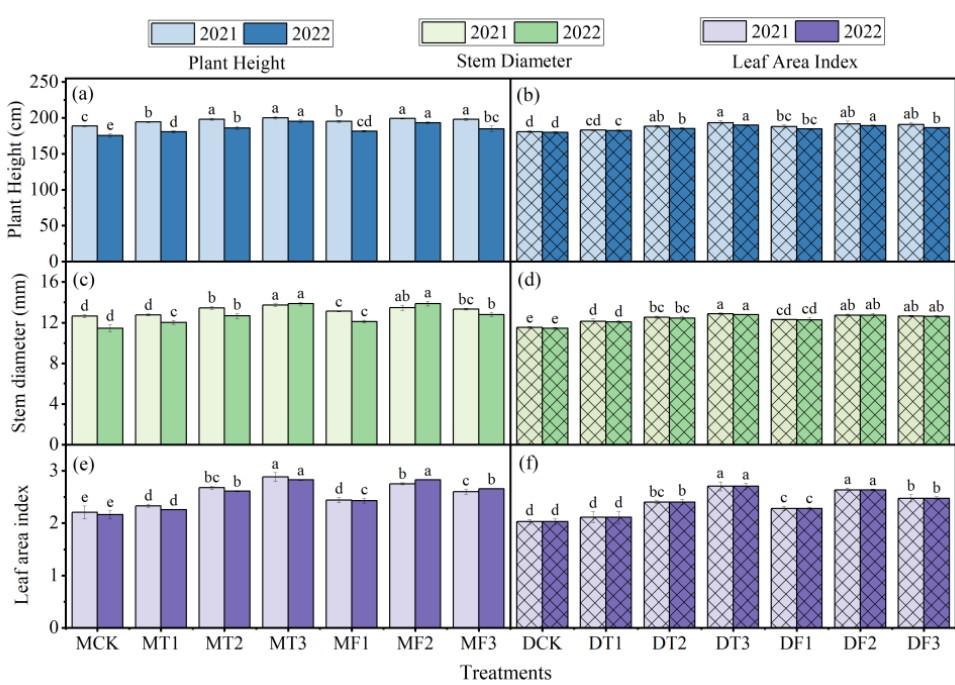

**Figure 2.** Tomato plant growth for different irrigation and fertilization treatments. Note: (**a**) is plant height of moistube irrigation in 2021 and 2022, (**b**) is plant height of drip irrigation in 2021 and 2022, (**c**) is stem diameter of moistube irrigation in 2021 and 2022, (**d**) is stem diameter of drip irrigation in 2021 and 2022, (**e**) is leaf area index of moistube irrigation in 2021 and 2022, (**f**) is leaf area index of drip irrigation in 2021 and 2022, Different lowercase letters in the figure indicate the significance of the differences between the treatments. $p < 0.05$.

### 3.2. Results of Tomato Fruit Quality

In this study, six quality indices of soluble sugar, titratable acid, vitamin C, sugar/acid ratio, lycopene, and nitrate content of tomatoes were determined, and the significance difference analysis of the six indices for different fertilization treatments was carried out, as shown in Figures 3 and 4. In addition to soluble sugar, there were significant differences in the contents of titratable acid, vitamin C, sugar/acid ratio, lycopene, and nitrate in the optimal values of MT3 and MF2 and DT3 and DF2 under Moistube irrigation and drip irrigation conditions. Using the two water-saving methods, the contents of soluble sugar, titratable acid, and nitrate increased with the increase in inorganic fertilizer concentration, and lycopene increased first and then decreased with the increase in inorganic fertilizer concentration. Compared with the control group, MT3 and DT3 obtained the maximum values of soluble sugar, titratable acid, nitrate, and lycopene in the treatment of inorganic fertilizer. The two-year average growth of MT3 was 11.80%, 24.41%, 47.78%, and 11.95%, respectively, and the two-year average growth of DT3 was 13.61%, 30.13%, 63.69%, and 14.19%, respectively. In addition, from the two-year average of soluble sugar, titratable acid, nitrate, and lycopene content, MT3 was 0.95%, −3.47%, −2.49%, and 5.03% more than DT3, respectively. With the increase in inorganic fertilizer concentration, the vitamin C content showed a trend of first increasing and then decreasing. MT2 and DT2 obtained the maximum value of vitamin C, with an average increase of 27.19% and 36.76% in the two years. In addition, MT2 increased by 12.76% more than the two-year average of DT2 vitamin C. The sugar/acid ratio decreased with the increase in inorganic fertilizer concentration. MT1 and DT1 obtained the maximum value of sugar/acid ratio, with a two-year average of 8.09 and 7.73, respectively, and MT1 increased by 4.66% more than DT1. The contents of soluble sugar, vitamin C, lycopene, and sugar/acid ratio increased first and then decreased with the increase in organic fertilizer concentration. The maximum values of soluble sugar, vitamin C, and lycopene were obtained by MF2 and DF2 treatments. The two-year average growth of MF2 was 12.92%, 27.73%, and 18.56%, respectively, and the two-year average

growth of DF2 was 14.49%, 45.04%, and 18.79%, respectively. In addition, from the two-year average of soluble sugar, vitamin C, and lycopene, MF2 increased by 0.74%, 6.69%, and 6.90% more than DF2, respectively. The sugar/acid ratio of MF2 and DF2 treatments was the largest, with an average value of 9.09 and 8.54, respectively, and MF2 increased by 6.44% compared with DF2. The content of titratable acid decreased with the increase in the concentration of organic fertilizer. MF3 and DF3 obtained the maximum value of titratable acid, with an average increase of 3.22% and 9.08% in the two years, and MF3 decreased by 4.76% compared with DF3. The nitrate content increased with the increase in organic fertilizer concentration. MF3 and DF3 obtained the maximum nitrate content, with an average increase of 35.36% and 46.91% in the two years, respectively, and MF3 decreased by 0.48% compared with DF3. In summary, with the same fertilization treatment, the contents of soluble sugar, sugar/acid ratio, vitamin C, and lycopene increased more than those of drip irrigation but less than those of drip irrigation.

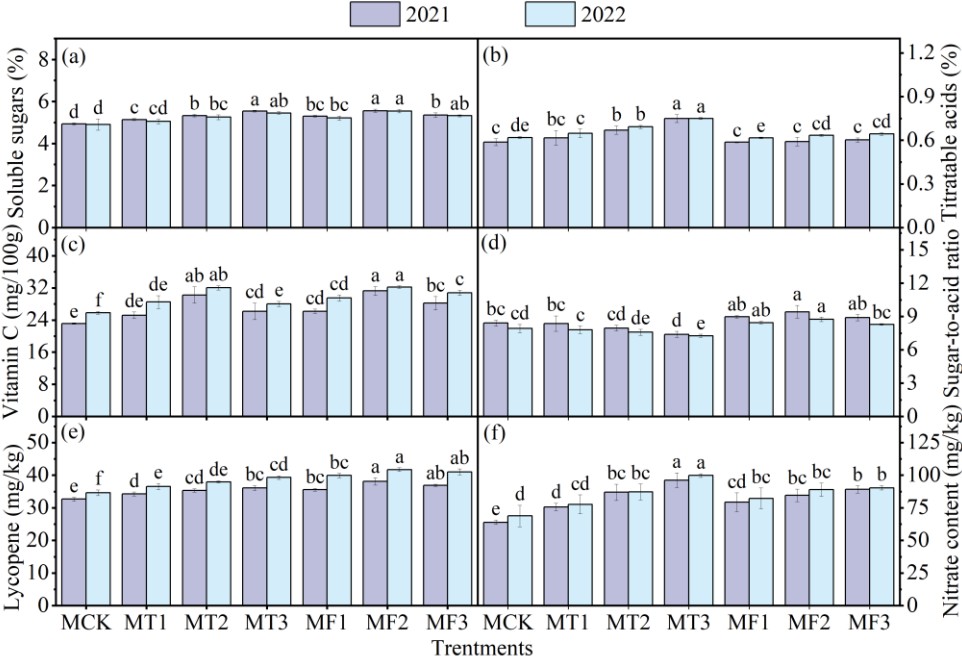

**Figure 3.** Tomato quality of different fertilization treatments under Moistube irrigation conditions. Note: (**a**) is soluble sugars, (**b**) is titratable acids, (**c**) is vitamin C, (**d**) is sugar-to-acid ratio, (**e**) is lycopene content, (**f**) is nitrate content. Different lowercase letters in the figure indicate the significance of the differences between the treatments. $p < 0.05$.

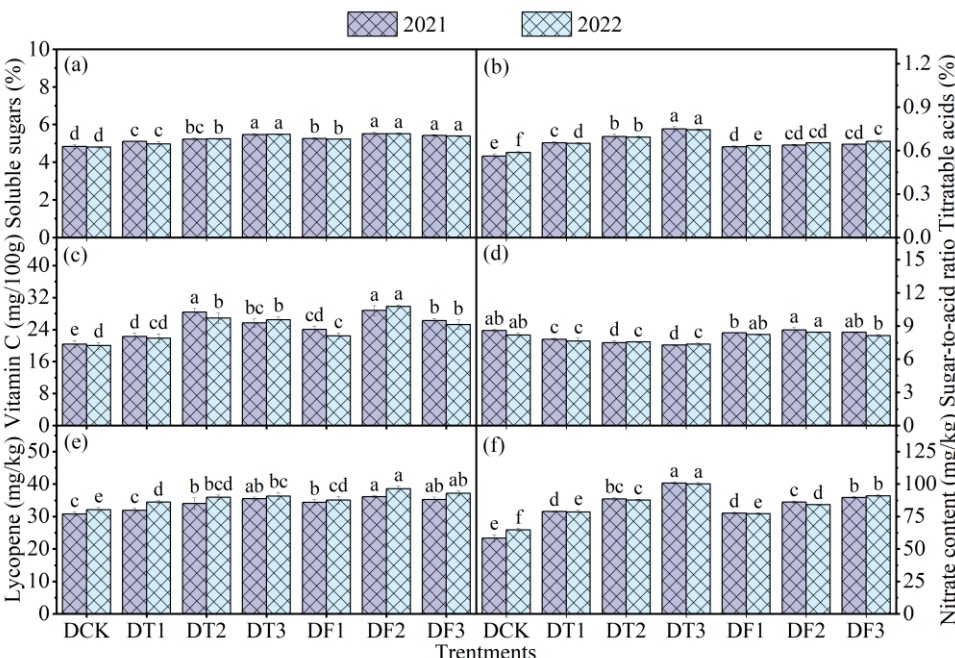

**Figure 4.** Tomato quality of different fertilization treatments under drip irrigation conditions. Note: (**a**) is soluble sugars, (**b**) is titratable acids, (**c**) is vitamin C, (**d**) is sugar-to-acid ratio, (**e**) is lycopene content, (**f**) is nitrate content. Different lowercase letters in the figure indicate the significance of the differences between the treatments. $p < 0.05$.

### 3.3. Results of Tomato Yield and Water and Fertilizer Efficiency

As can be seen in Figure 5, the results of the two-year experiment showed that there was no significant difference in total yield and fruit weight between MT3 and DT3 and MF2 and DF2 with the same irrigation method, but there was a significant increase compared to the control group. Using the modes of Moistube irrigation and drip irrigation, the total yield and fruit weight of tomatoes increased with the increase in inorganic fertilizer concentration, and it revealed a pattern where the amount of soluble organic fertilizer increased at first then decreased. The maximum total yield and fruit weight were MT3 and DT3 treatments for all fertilization treatments. The two-year average growth of total yield and single fruit weight of MT3 was 32.72% and 34.85%, respectively, and the two-year average growth of total yield and single fruit weight of DT3 was 31.38% and 27.49%, respectively. The two-year average increase in total yield and single fruit weight of MF2 was 27.85% and 34.40%, respectively, and the two-year average increase in total yield and single fruit weight of DF2 was 28.50% and 26.49%, respectively. In addition, using the same fertilization treatment, the two-year average of total yield and single fruit weight of MT3 increased by 0.56% and 4.84% compared to DT3, respectively, and the two-year average of total yield and single fruit weight of MF2 increased by 0.49% and 5.32% compared to DF2, respectively. The results indicated that the yield increase effect of Moistube irrigation was better than that of drip irrigation. According to Figure 6, there were significant differences in PFP and WUE between all fertilization treatments using the water-saving methods of Moistube irrigation and drip irrigation. Using the two irrigation methods, the partial fertilizer productivity decreased with the increase in inorganic fertilizer and organic fertilizer concentration, the values of MT1 and DT1 treatments were the largest, and the values of MT3 and DT3 treatments were the smallest. According to the two-year average calculation of PFP, the difference between MT1 and MT3 is 151.50%, and the difference between DT1 and DT3 is 156.35%. The concentration of inorganic fertilizer led to a rise in water use efficiency, while the concentration of soluble organic fertilizer showed a tendency to initially increase and then drop. The best performance was exhibited by the MT3 and DT3 treatments, followed by MF2 and DF2 treatments. According to the two-year average calculation of WUE, the

difference between MT3 and MF2 is only 0.31, and the difference between DT3 and DF2 is only 0.51.

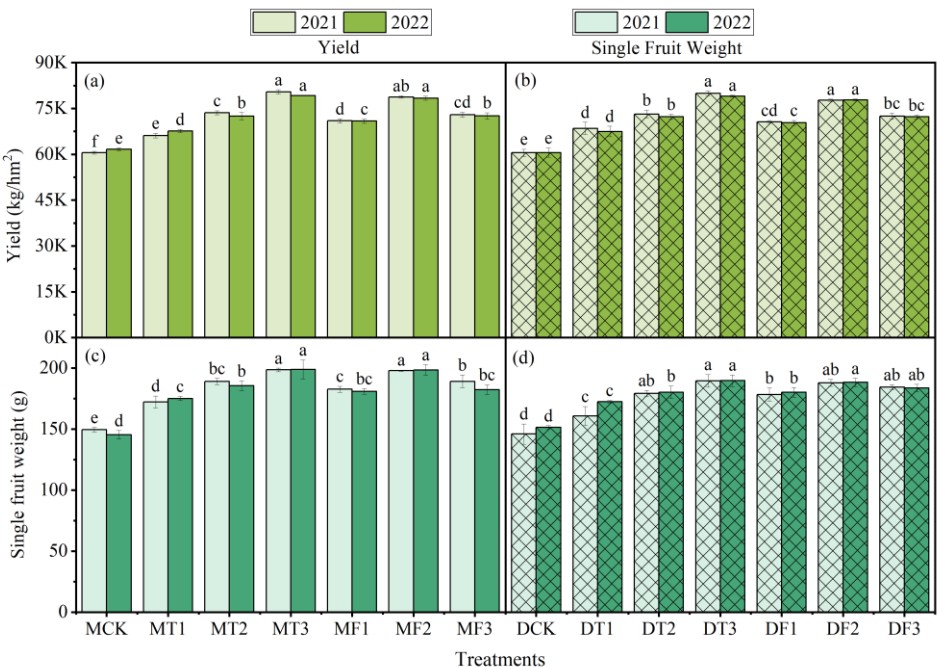

**Figure 5.** Tomato yield and fruit weight for different irrigation and fertilization treatments. Note: (**a**) is yield of moistube irrigation in 2021 and 2022, (**b**) is yield of drip irrigation in 2021 and 2022, (**c**) is single fruit weight of moistube irrigation in 2021 and 2022, (**d**) is single fruit weight of drip irrigation in 2021 and 2022. Different lowercase letters in the figure indicate the significance of the differences between the treatments. $p < 0.05$.

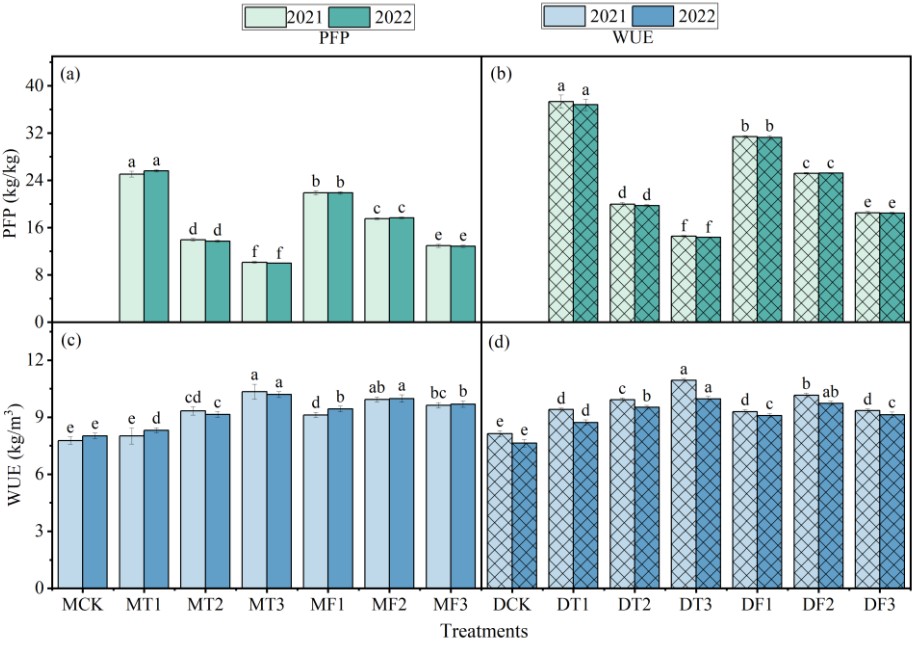

**Figure 6.** Partial fertilizer productivity and water use efficiency for different irrigation and fertilization treatments. Note: PFP is partial fertilizer productivity, WUE is water use efficiency. (**a**) is PFP of moistube irrigation in 2021 and 2022, (**b**) is PFP of drip irrigation in 2021 and 2022, (**c**) is WUE of moistube irrigation in 2021 and 2022, (**d**) is WUE of drip irrigation in 2021 and 2022. Different lowercase letters in the figure indicate the significance of the differences between the treatments. $p < 0.05$.

### 3.4. Comprehensive Evaluation of Tomatoes Based on the Entropy-Weight TOPSIS Model

In this study, four categories of tomatoes were constructed, including growth index, quality index, yield index, and efficiency index. The entropy-weight method was used to calculate the weights of 13 indicators in the above four categories, and the results are shown in Table 5. In 2021, the maximum weight was PFP at 9.639%, followed by the sugar/acid ratio at 9.547%, and the minimum weight was the stem diameter at 5.539%. In 2022, the maximum weight was yield at 10.089%, followed by PFP at 9.718%, and the minimum weight at 5.430%.

**Table 5.** Weighting results of each indicator in 2021 and 2022.

| Index | 2021 | | | 2022 | | |
|---|---|---|---|---|---|---|
| | $e_i$ | $d_i$ | Weight | $e_i$ | $d_i$ | Weight |
| C1 | 0.937 | 0.063 | 6.999 | 0.938 | 0.062 | 6.552 |
| C2 | 0.950 | 0.050 | 5.539 | 0.932 | 0.068 | 7.187 |
| C3 | 0.926 | 0.074 | 8.213 | 0.915 | 0.085 | 8.925 |
| C4 | 0.942 | 0.058 | 6.368 | 0.934 | 0.066 | 6.992 |
| C5 | 0.926 | 0.074 | 8.205 | 0.935 | 0.065 | 6.839 |
| C6 | 0.938 | 0.062 | 6.825 | 0.931 | 0.069 | 7.261 |
| C7 | 0.913 | 0.087 | 9.547 | 0.914 | 0.086 | 9.073 |
| C8 | 0.943 | 0.057 | 6.289 | 0.938 | 0.062 | 6.478 |
| C9 | 0.923 | 0.077 | 8.507 | 0.913 | 0.087 | 9.200 |
| C10 | 0.921 | 0.079 | 8.668 | 0.904 | 0.096 | 10.089 |
| C11 | 0.937 | 0.063 | 6.985 | 0.948 | 0.052 | 5.430 |
| C12 | 0.913 | 0.087 | 9.639 | 0.908 | 0.092 | 9.718 |
| C13 | 0.926 | 0.074 | 8.215 | 0.941 | 0.059 | 6.256 |

Note: $e_i$ is information entropy, $d_i$ is utility value. C1: plant height, C2: stem diameter, C3: leaf area index, C4: soluble sugars, C5: titratable acids, C6: vitamin C, C7: sugar-to-acid ratio, C8: lycopene, C9: nitrate content, C10: yield, C11: single fruit weight, C12: partial fertilizer productivity, C13: water use efficiency.

According to the objective weights of each index determined with the entropy-weight method, the positive and negative ideal solutions of the 14 evaluation objects were calculated. The results of the ideal solution for the two-year test are shown in Table 6.

**Table 6.** Ideal solution distances for each treatment in 2021 and 2022.

| Treatment | 2021 | | 2022 | |
|---|---|---|---|---|
| | Positive Ideal Solution | Negative Ideal Solution | Positive Ideal Solution | Negative Ideal Solution |
| MCK | 0.748 | 0.437 | 0.793 | 0.400 |
| MT1 | 0.550 | 0.506 | 0.588 | 0.458 |
| MT2 | 0.456 | 0.646 | 0.504 | 0.576 |
| MT3 | 0.558 | 0.733 | 0.571 | 0.750 |
| MF1 | 0.403 | 0.633 | 0.455 | 0.603 |
| MF2 | 0.283 | 0.839 | 0.289 | 0.867 |
| MF3 | 0.390 | 0.686 | 0.435 | 0.643 |
| DCK | 0.857 | 0.452 | 0.848 | 0.452 |
| DT1 | 0.669 | 0.454 | 0.653 | 0.469 |
| DT2 | 0.565 | 0.498 | 0.548 | 0.490 |
| DT3 | 0.590 | 0.657 | 0.586 | 0.628 |
| DF1 | 0.503 | 0.541 | 0.520 | 0.553 |
| DF2 | 0.350 | 0.697 | 0.327 | 0.721 |
| DF3 | 0.472 | 0.557 | 0.485 | 0.550 |

Note: CK is no fertilization; T1, T2, and T3 are low-, medium-, and high-concentration inorganic fertilizers, respectively; and F1, F2, and F3 are the combinations of low-, medium-, and high-concentration soluble organic fertilizers and low-concentration inorganic fertilizers. M and D stand for Moistube irrigation and drip irrigation, respectively.

According to the ideal solution values of 14 evaluation objects of Moistube irrigation and drip irrigation, the Euclidean distance between each evaluation object and the positive ideal solution and negative ideal solution was calculated. Then, according to the Euclidean distance calculation, the closeness was obtained, and finally the scores, of the comprehensive TOPSIS evaluation of the entropy-weight method were obtained and ranked. The comprehensive scores and rankings for the two years are shown in Table 7. In both 2021 and 2022, MF2 ranked first, followed by DF2. The overall ranking ordinal difference obtained from the two-year trial was 2: MCK, MT1, and MT2 in 2022 decreased by 1 place compared with the 2021 ranking; MT3, DCK, and DT2 in 2022 each increased by 1 place compared with the 2021 ranking; and the rankings of other treatments did not change. In 2021, the maximum value was 0.758, the minimum value was 0.345, and the standard deviation of the score was 0.110. The maximum value for 2022 was 0.750, the minimum value was 0.348, and the standard deviation of the score was 0.112. In summary, it can be concluded that the combination of low-concentration inorganic fertilizer and medium-concentration soluble organic fertilizer is the best fertilization treatment, and the Moistube irrigation method is more suitable for the cultivation of greenhouse tomatoes in the Loess Plateau than drip irrigation. At the same time, the two-year comprehensive ranking showed that the entropy-weight TOPSIS model adopted in this study had little fluctuation in the comprehensive evaluation of tomatoes, and the results of selecting the best fertilization combination were relatively stable.

**Table 7.** Comprehensive score and ranking of the entropy-weight TOPSIS model.

| Treatment | 2021 | | 2022 | | Ranking Change |
|---|---|---|---|---|---|
| | Score | Rank | Score | Rank | |
| MCK | 0.369 | 13 | 0.335 | 14 | ↓ |
| MT1 | 0.479 | 10 | 0.438 | 11 | ↓ |
| MT2 | 0.586 | 5 | 0.533 | 6 | ↓ |
| MT3 | 0.568 | 6 | 0.568 | 5 | ↑ |
| MF1 | 0.611 | 4 | 0.570 | 4 | – |
| MF2 | 0.758 | 1 | 0.750 | 1 | – |
| MF3 | 0.637 | 3 | 0.597 | 3 | – |
| DCK | 0.345 | 14 | 0.348 | 13 | ↑ |
| DT1 | 0.404 | 12 | 0.418 | 12 | – |
| DT2 | 0.468 | 11 | 0.472 | 10 | ↑ |
| DT3 | 0.523 | 8 | 0.517 | 8 | – |
| DF1 | 0.518 | 9 | 0.516 | 9 | – |
| DF2 | 0.666 | 2 | 0.688 | 2 | – |
| DF3 | 0.541 | 7 | 0.532 | 7 | – |

Note: CK is no fertilization; T1, T2, and T3 are low-, medium-, and high-concentration inorganic fertilizers, respectively; and F1, F2, and F3 are the combinations of low-, medium-, and high-concentration soluble organic fertilizers and low-concentration inorganic fertilizers. M and D stand for Moistube irrigation and drip irrigation, respectively. ↓ represents the decline, ↑ represents the rise, – represents no change.

## 4. Discussion

### 4.1. Effects of Different Treatments on Tomato Fruit Quality

The concentrations of various quality indices of tomato fruit differed among varying irrigation methods and fertilization treatments. Soluble sugar and titratable acid content are the intrinsic quality indicators that characterize the taste of tomatoes; the larger the sugar/acid ratio, the better the flavor, and vice versa. In this study, compared with the application of inorganic fertilizer alone, the sweetness of tomatoes with soluble organic fertilizer was greatly improved using the two irrigation methods. Although the acidity was also increased, it was reduced by 21.05% compared with that of the inorganic fertilizer treatment. Therefore, the tomato treated with soluble organic fertilizer had a better taste. This was consistent with the conclusion of Li et al. [34] that organic fertilizer can greatly increase the sweetness of tomatoes and improve their taste. In this study, the nitrate content of both irrigation methods increased with an increase in the concentration of inorganic

fertilizer and soluble organic fertilizer. At the same time, the results of the two-year test showed that the nitrate content of the combined application of soluble organic fertilizer was lower than that of the inorganic fertilizer treatment. This was consistent with the experimental conclusions of Zhao et al. [35] and Wu et al. [16] that organic fertilizer can significantly reduce the nitrate content in tomato fruits. With different irrigation methods, the contents of vitamin C and lycopene showed a parabolic change with the increase in the concentration of inorganic fertilizer and soluble organic fertilizer. Moreover, the contents of VC and lycopene with the combined application of soluble organic fertilizer were generally higher than those with the treatment of inorganic fertilizer. This was consistent with the experimental conclusions of Fracchiolla et al. [36] and Guo et al. [37] that soluble organic fertilizer can significantly increase the nutrient content of tomato fruits. Combined with the application of soluble organic fertilizer, the content of soluble sugar, sugar/acid ratio, vitamin C, and lycopene showed that the content of the Moistube irrigation method was higher than that of drip irrigation. This was similar to the experimental conclusions of Liu et al. [24] that Moistube irrigation takes advantage of its continuous and stable supply of sufficient water for the root zone of crops, which greatly improves the quality of crops.

### 4.2. Effects of Different Treatments on Tomato Growth, Yield, and Efficiency

For different irrigation methods, the growth indices, yield indices, and water use efficiency of tomatoes increased with the increase in inorganic fertilizer concentration. The results of Xing et al. [38] and Zhu et al. [39] were different, and the results showed that increasing the amount of fertilizer had a positive effect on tomato growth, yield, and water use efficiency and had a negative effect on tomato growth beyond a certain range. This may be the inflection point for the positive and negative effects of the fertilizer gradient setting in this experiment to reach a small span. With the increase in soluble organic fertilizer concentration, it showed a trend of first increasing and then decreasing. This is consistent with the experimental conclusions of Li et al. [40], Li et al. [41], and Wu et al. [16]. The appropriate amount of soluble organic fertilizer can increase soil nutrients and enhance the root vitality of crops so as to effectively promote plant growth, improve tomato yield, and increase water use efficiency [13,42,43]. Moistube irrigation increases yield more than drip irrigation and has higher water use efficiency. This is similar to the experimental conclusions of Liu et al. [24]. With different irrigation methods, the partial fertilizer productivity decreased with the increase in inorganic fertilizer and organic fertilizer concentrations. This is consistent with the experimental conclusions of Shang et al. [44] and Zhang et al. [45]. This may be due to the increase in yield being lower than that of soil fertilizer, so the PFP of tomatoes decreased with the increase in fertilizer amount.

### 4.3. A Comprehensive Evaluation Model of Tomatoes

By calculating and ranking the closeness of the 14 evaluation objects, the composite scores of both years showed that the best treatment was MF2, followed by DF2. The comprehensive evaluation conclusions of the two-year experiment using the entropy-weight TOPSIS model are consistent, indicating that the results of the method are stable and reliable. Sun et al. [46] studied the effects of water saving and nitrogen reduction on the quality, yield, water yield, and nitrogen use efficiency of tomatoes. The entropy-weight TOPSIS model was also used to optimize the water and nitrogen management of greenhouse tomatoes, and the best irrigation and nitrogen application rates were selected. Zheng et al. [47] explored the effects of biogas slurry on tomatoes at different growth stages and used the entropy-weight TOPSIS model to comprehensively evaluate the key stages of biogas slurry on the formation of tomato yield and quality. Zhang et al. [48] used the entropy-weighted TOPSIS model to explore the optimal combination of drip irrigation and fertilization rates for different potato varieties. Many researchers have employed this technique to conduct relevant research, demonstrating the benefits of the entropy-weight TOPSIS model's excellent adaptability, broad applicability, and consistent outcomes. As a result, the evaluation outcomes produced by applying this technique are trustworthy.

## 5. Conclusions

In this study, the combined application mode of organic fertilizer and inorganic fertilizer using different irrigation methods was discussed, and a suitable combination of irrigation and fertilization for greenhouse tomatoes in the Loess Plateau was found. The results of the two-year experiment show that the growth characteristics of tomato plants were the best with the treatment of low-concentration inorganic fertilizer combined with medium-concentration soluble organic fertilizer, regardless of whether Moistube irrigation or drip irrigation was used. Compared with the application of inorganic fertilizer alone, the application of soluble organic fertilizer not only increased the yield of tomatoes but also improved fruit quality and taste. Different fertilization treatments had significant effects on the fertilizer partial productivity and water production efficiency of tomatoes. The entropy-weight TOPSIS model was used to comprehensively evaluate different fertilization treatments with the two irrigation methods, and it was found that MF2 was the optimal treatment. Tomatoes with low-concentration inorganic fertilizer combined with medium-concentration soluble organic fertilizer under Moistube irrigation conditions had the best comprehensive performance in four aspects: growth characteristics, fruit quality, yield, and efficiency.

**Author Contributions:** B.L. edited the draft; L.S. reviewed and edited the draft. All authors have read and agreed to the published version of the manuscript.

**Funding:** This research was supported by the Basic Research Program of Shanxi Province (202103021224093) and the Natural Science Foundation of Shanxi Province (201801D121266, 201901D111059).

**Institutional Review Board Statement:** Not applicable.

**Data Availability Statement:** The data that support the findings of this study are available from the corresponding author upon reasonable request.

**Conflicts of Interest:** The authors declare no conflicts of interest.

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
