# Peer review of "Effects of Soluble Organic Fertilizer Combined with Inorganic Fertilizer on Greenhouse Tomatoes with Different Irrigation Techniques"

_agriculture, doi:10.3390/agriculture14020313_

Round 1

Reviewer 1 Report

Comments and Suggestions for Authors

There is too much repetition of similar infromation in different sections

Avoid writing very long sentences, they are diffiuclt to comprehend for readers

In the discussion section, there is more of repetition of results interpretation and very limited results discussion. Revise the section

There is no conclusion in the conclusion section but more of outlining of results. Redo the section

Comments on the Quality of English Language

Reviewer 2 Report

Comments and Suggestions for Authors

The work demonstrates effort on the part of the authors; however, it presents several aspects that need to be explained, improved, or modified before being considered for publication, such as:

The authors consistently mention a two-year trial, but the methodology only indicates that it took place between May and October 2021 (Ln 117).

The authors do not provide details about the plant material used, which is a relevant factor for the results. They only mention the variety (Ln 125), but why that specific one?

Tables and figures should be self-explanatory and provide details for all acronyms without the need to refer to the text.

The use of a nutrient-free control solution is not representative, as real and/or commercial conditions do not involve such absence of nutrients. Not providing nutrients to the control solution may distort the obtained results since it is not a common practice and puts it at a disadvantage compared to other trials that do apply nutrients.

The methodology for tomato quality parameters is vague and needs improvement.

The discussion is lacking and has minimal engagement with other studies or different varieties.

Comments on the Quality of English Language

Nd

Round 2

Reviewer 1 Report

Comments and Suggestions for Authors

Comments on the Quality of English Language

Author Response

Line335,(the difference significance analysis)Correct this to ..the significance difference analysis

Answer: It has been changed to the significance difference analysis. (line337-338)

Line460, (caters for it) Delete this

Answer: It has been deleted. (line464)

Line480 (4.2 effect of tomato growth, and efficiency) Effect of what? There is a word or information missing between of and tomato growth

Answer: Modifications have been made.

4.1 Effects of different treatments on tomato fruit quality (line460)

4.2 Effects of different treatments on tomato growth, yield, and efficiency (line489)

Reviewer 2 Report

Comments and Suggestions for Authors

The authors have incorporated nearly all the suggested modifications; however, there is still one point that needs improvement. The methodology should provide references for the methods used, giving details of the equipment employed. The discussion remains vague (although improvements are evident since the first version).

Comments on the Quality of English Language

If the authors modify the methodology, I believe the work could be suitable for publication. In its current form, I would discard it, as it does not provide all the information necessary to ensure replicability and objectivity.

Author Response

Comments and suggestions for authors

The authors have incorporated nearly all the suggested modifications;however,there is still one point that needs improvement. The methodology should provide references for the methods used, giving details of the equipment employed. The discussion remains vague (although improvements are evident since the first version).

Answer: References have been provided for all methods used, see Ref. 26-31. The details of the equipment employed is supplemented. The discussion section has also been revised again, as detailed in the yellow text of the discussion section.

Comments on the quality of English language

If the authors modify the methodology, I believe the work could be suitable for publication. In its current form, I would discard it, as it does not provide all the information necessary to ensure replicability and objectivity.

Answer: For the quality of English, the author service in the MDPI system has been adopted to comprehensively rewrite the English language.